# Incentive and Dynamic Client Selection for Federated Unlearning

Submission Id: 872

## ABSTRACT

With the development of AI-Generated Content (AIGC), data is becoming increasingly important, while the right of data to be forgotten, which is defined in the General Data Protection Regulation (GDPR) and permits data owners to remove information from AIGC models, is also arising. To protect this right in a distributed manner corresponding to federated learning, federated unlearning is employed to eliminate history model updates and unlearn the global model to mitigate data effects from the targeted clients intending to withdraw from training tasks. To diminish centralization failures, the hierarchical federated framework that is distributed and collaborative can be integrated into the unlearning process, wherein each cluster can support multiple AIGC tasks. However, two issues remain unexplored in current federated unlearning solutions: 1) getting remaining clients, those not withdraw from the task, to join the unlearning process, which demands additional resources and notably has fewer benefits than federated learning, particularly in achieving the original performance via alternative unlearning processes and 2) exploring mechanisms for dynamic unlearning in the selection of remaining clients possessing unbalanced data to avoid starting the unlearning from scratch. We initially consider a two-level incentive and unlearning mechanism to address the aforementioned challenges. At the lower level, we utilize evolutionary game theory to model the dynamic participation process, aiming to attract remaining clients to participate in retraining tasks. At the upper level, we integrate deep reinforcement learning into federated unlearning to dynamically select remaining clients to join the unlearning process to mitigate the bias introduced by the unbalanced data distribution among clients. Experimental results demonstrate that the proposed mechanisms outperform comparative methods, enhancing utilities and improving accuracy.

## CCS CONCEPTS

• **Networks** → *Network privacy and anonymity*.

## KEYWORDS

federated unlearning, dynamic retraining, deep reinforcement learning

## 1 INTRODUCTION

The advent of ChatGPT, an Artificial Intelligence (AI)-powered chatbot developed by OpenAI [1], has precipitated a surge in attention towards AI-Generated Content (AIGC) from both industry and academia. Substantial data, derived from the public internet or personal information, is employed to train AIGC models, enhancing user experience. Consequently, data is being considered as the oil in the burgeoning AI era. To protect data ownership and the right to be forgotten, defined in the General Data Protection Regulation (GDPR), which has recently grasped significant attention, individuals are permitted to remove their private data from well-trained AIGC models. Machine unlearning [2] is proposed to eliminate data influence from a trained model without necessitating retraining from scratch, ensuring that the right to be forgotten is safeguarded. Specifically, considering that there are targeted samples that targeted clients intending to withdraw from training tasks require to be removed from the well-trained model, the model should undergo an unlearning process to ensure it performs as if it has never encountered the target samples.

Federated unlearning is one scenario within machine unlearning and represents a means of unlearning within Federated Learning (FL). Unlike conventional methods that upload data directly to central servers, FL requires clients to share model updates with central servers to train a global model, thereby protecting the privacy of raw data [3]. It is impractical to eliminate data samples from an FL-trained model by training a new model from scratch using datasets from remaining clients, which do not withdraw data effects from the well-trained FL model. Therefore, federated unlearning should also engage in the removal of targeted historical model updates to eliminate the data effects of certain clients, a process distinct from that of machine unlearning [4]. For instance, assuming there are targeted clients that plan to eliminate data effects from a trained FL model, the global model should first remove historical model updates from those targeted clients and subsequently unlearn for a few rounds with the remaining clients to mitigate the data effects. This approach has been employed in FedEraser [4], RapidRetrain [5], and FedRecovery [6].

However, current federated unlearning solutions have not addressed the following issues. First, the FL-trained model necessitates the participation of remaining clients in the unlearning process to diminish the data effects of targeted clients. This unlearning process imposes an additional burden by requiring the computational and communication resources of the remaining clients. Second, selecting remaining clients to participate in the unlearning process presents a significant challenge due to the potential presence of unbalanced non-independent and Identically Distributed (non-IID) data among those attractive remaining clients, which may subsequently diminish the performance of the unlearning FL models. Present solutions predominantly utilize a random selection method, neglecting to data imbalances among remaining clients, which is unsuitable for dynamic environments.

In this paper, we introduce a two-tiered incentive and unlearning mechanism to address the aforementioned issues. At the lower level, we design an incentive mechanism based on evolutionary game theory to motivate the remaining clients to participate in the unlearning process. At the upper level, we propose a method that integrates deep reinforcement learning (DRL) with federated unlearning, with the aim of selecting remaining clients possessing unbalanced local data dynamically. The principal contributions of this paper are articulated as follows:

- We propose a cluster-based joint incentive and unlearning framework for federated unlearning. The cluster is utilized to perform multiple types of federated unlearning tasks

and reduce the impact on remaining clients. We discern a correlation between the data distribution of local datasets from the remaining clients and the performance of the unlearning models.

- We model incentive decisions of the remaining clients using an evolutionary game, aiming to capture their dynamics and rationality, and attract them to join the unlearning process, which demands additional resources.
- To achieve dynamic federated unlearning, based on the evolutionary game, we integrate DRL with federated unlearning to choose remaining clients and find dynamic strategies to mitigate the effects of unbalanced local data.

## 2 RELATED WORK

*Client Federated Unlearning.* The application of machine unlearning to federated unlearning was first proposed in FedEraser [4], wherein the authors leverage the central server's storage to archive historical model updates, employing them to unlearn the FL models and thereby reduce unlearning time. The remaining clients and the central server collaboratively utilize historical model updates to execute a process of local calibrating, updating, aggregating, and unlearning model updates, thereby calibrating the trained FL model and eliminating the data influence of the target clients. [5] proposed an Adahessian-based rapid retraining method to reduce computational costs. [6] employed differential privacy in the historical model updates of federated unlearning to unlearn a global model that is irrelevant to the data contributions of targeted clients. [7] introduced a clustered aggregation-based asynchronous federated unlearning approach to minimize affected clients and formulated a lexicographic minimization problem to optimize client-cluster assignment. [8] first highlighted the risks of inference information leakage in federated unlearning and outlined potential defense approaches to safeguard client privacy. [9] employed knowledge distillation, using teacher and student networks, to extract knowledge from the historical model updates of targeted clients and to train the unlearned global model, thereby enhancing model performance. [10] introduced a federated unlearning method, utilizing projected gradient ascent to formulate the unlearning process as a constrained maximization problem to maximize the loss of the local model for unlearning while preserving the knowledge from the remaining clients. However, existing client federated unlearning methods select random or predefined remaining clients to unlearn the global model, without considering the impact of unbalanced local data on remaining clients and the adaptability for dynamic environments.

*Sample and Class Federated Unlearning.* [9] integrated reverse stochastic gradient ascent with elastic weight consolidation in federated unlearning to eliminate the influence of the targeted training data. [11] introduced an FL-based unlearning framework, designed for digital twin mobile networks, that incorporates memory evaluation and erase modules. This framework integrates key feature maps to obtain memory evaluation information and employs a multi-loss training method to unlearn data while enhancing accuracy. [12] introduced a Bayesian federated unlearning approach, employing a parameter self-sharing method to navigate the trade-off between

forgetting data from the targeted client and maintaining the accuracy of the global model. [13] developed a data valuation method based on shared Shapley values for model markets, designed to evaluate data value following the unlearning of data by targeted clients. [14] employed the Term Frequency-Inverse Document Frequency (TF-IDF) method to assess the influence between channels and classes in image classification, pruning the most pertinent channel to unlearn the contribution of a specific class. Since the aforementioned methods engage in the process of model updates, the sample and class-federated unlearning methods can be incorporated into both the existing client-federated unlearning methods and the proposed method to enhance model performance.

## 3 PRELIMINARIES

### 3.1 Federated Learning

Federated learning, initially introduced by [3], necessitates multiple clients to share model updates, rather than raw data, with a central server for training and aggregation, thereby safeguarding the data privacy of the clients. Given $\mathcal{N} = \{1, \ldots, n, \ldots, N\}$ of $N$ clients and a central server $\mathcal{J} = \{1\}$, and defining the global loss function with model updates $\mathbf{w}^k$ at the $k$th round as $F(\mathbf{w}^k)$. As shown in Figure 1 (a), the collaborative workflow of federated learning is iteratively repeated over $K$ training rounds as follows:

**0) Model Distribution.** Initially, the central server replays the global model updates $\mathbf{w}^k$ to clients, ensuring that each participating device starts the local training process with the most recent model parameters.

**1) Local Computation.** Upon receiving $\mathbf{w}^k$, each client $n \in \mathcal{N}$ proceeds to compute the local model updates $\mathbf{w}_n^k$ using their respective local datasets $D_n$, without sharing raw data with the server or other clients.

**2) Update Transmission.** Each client $n$ transmits the updated local model updates $\mathbf{w}_n^k$ to the central server for subsequent aggregations.

**3) Global Aggregation.** Upon receiving the updated model updates $\mathbf{w}_n^k$ from each client in $\mathcal{N} = \{1, \ldots, n, \ldots, N\}$, the central server utilizes the FedAvg algorithm [3], denoted as $\mathbf{w}^{k+1} = \frac{1}{N} \sum_{n=1}^{N} \mathbf{w}_n^k$, to aggregate them, yielding the updated global model $\mathbf{w}^{k+1}$. Subsequently, it broadcasts $\mathbf{w}^{k+1}$ to clients for the $(k+1)$th round.

To address issues related to centralization and communication overheads, clustered FL has been widely adopted, wherein multiple central servers serve as cluster heads, denoted as $\mathcal{J} = \{1, \ldots, j, \ldots, J\}$. Each client $n$ is associated with a cluster head $j$ to train FL models collaboratively. The training process within each cluster adheres to the aforementioned methodology.

### 3.2 Federated Unlearning

Federated unlearning was first introduced in [4], providing a mechanism that leverages the storage of model updates to enable the global model to forget the data effects originating from the targeted unlearning clients. Consider a scenario where a target client $n_u$ triggers an unlearning request to remove data effects from the global model $\mathbf{w}^k$ at $k$th training round. As shown in Figure 1 (b),

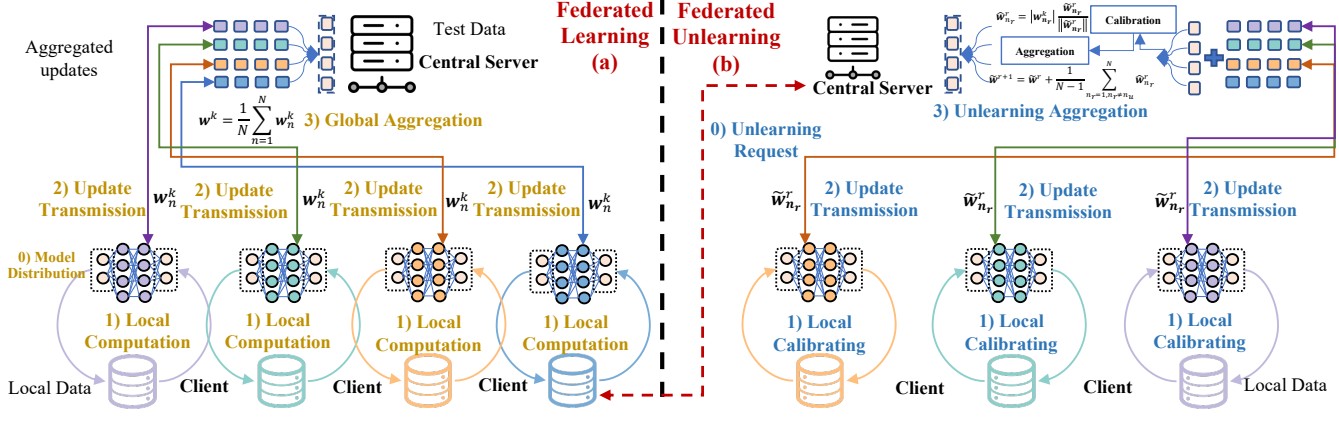

**Figure 1: Federated learning and unlearning**

the workflow of federated unlearning is then iteratively repeated over $R$ unlearning rounds, as follows:

**0) Unlearning Request.** Initially, upon receiving an unlearning request from a target client $n_u \in \mathcal{N}_u$, the central server extracts and removes the historical model updates pertinent to client $n_u$. Subsequently, it broadcasts the most recent global model without the updates from $n_u$ to the remaining clients, initiating the $r$th unlearning round.

**1) Local Calibrating.** At the $r$th unlearning round, each remaining client $n_r \in \mathcal{N} \backslash n_u$ actively engages in computing the local model updates $\widetilde{\mathbf{w}}_{n_r}^r$ using their respective local data $D_{n_r}$ for calibration, aiming to refine and adjust the model in the absence of the contributions from the targeted client $n_u$. To mitigate the computational costs associated with local retraining, FedEraser [4] reduces the number of local calibrating rounds compared with the original training process.

**2) Update Transmission.** Each remaining client $n_r \in \mathcal{N} \backslash n_u$ communicates the calibrated local model updates $\widetilde{\mathbf{w}}_{n_r}^r$ to the central server for subsequent aggregations.

**3) Unlearning Aggregation.** The central server collects the calibrated model updates $\widetilde{\mathbf{w}}_{n_r}^r$ from each remaining client within $\mathcal{N} \backslash n_u$. It utilizes $\widetilde{\mathbf{w}}_{n_r}^r$ to calibrate historical model updates $\mathbf{w}_{n_r}^r$ via the step length and direction layer by layer, which can be denoted as $\hat{\mathbf{w}}_{n_r}^r = |\mathbf{w}_{n_r}^r| \frac{\widetilde{\mathbf{w}}_{n_r}^r}{||\widetilde{\mathbf{w}}_{n_r}^r||}$. The central server then aggregates these parameter calibrations via $\widetilde{\mathbf{w}}^{r+1} = \widetilde{\mathbf{w}}^r + \frac{1}{N-1} \sum_{n_r=1, n_r \neq n_u}^N \hat{\mathbf{w}}_{n_r}^r$, effectively unlearning contributions from the targeted client $n_u$. Subsequently, the aggregated calibration model updates $\widetilde{\mathbf{w}}^{r+1}$ are broadcast to the remaining clients $\mathcal{N} \backslash n_u$ for the $(r+1)$th unlearning round.

## 3.3 Challenges

Current federated unlearning methods often overlook the impact on remaining clients, who are required to collaboratively unlearn the global model using additional computational and communication resources. Furthermore, these methods conventionally assume a random or predefined selection of remaining clients, neglecting the potential impact of unbalanced local data among these participants. Instead, as shown in Figure 2, we focus our study on a two-level

incentive and unlearning mechanism as follows: i) in the lower level, we formulate an evolutionary game to attract remaining clients to join the unlearning process and maximize the utilities, and ii) in the upper level, we utilize DRL based client selection mechanism to dynamically choose a subset of remaining clients to reduce data bias.

## 4 LOWER-LEVEL EVOLUTIONARY GAME

### 4.1 Game Formulation

As illustrated in Figure 2, we formulate the participation of the remaining clients in unlearning tasks as an evolutionary game, which elucidates the dynamics of clients transitioning between tasks to maximize their utilities.

*Players.* The remaining clients $\mathcal{N}_r\{1, \ldots, n_r, \ldots, N_r\} = \mathcal{N} \backslash N_u$ serve as the $N_r$ players of the evolutionary game, each determining different strategies for associating with central servers.

*Populations.* The clients are divided into a set $\mathcal{P} = \{1, \ldots, p, \ldots, P\}$ consisting of $P$ populations based on data quantities $D_p$ where $\cup_{p \in \mathcal{P}} |p| = N_r$. Each remaining client $n_r$ within a population $p$ possesses identical data samples $d_p$ [15].

*Strategy.* Each client $n_r$ within a population $p$ selects a central server $j$ to participate in the unlearning process with the aim of obtaining rewards and maximizing their utilities. This decision-making can be represented as $\mathcal{S}_{n_r}^p = \{a_{n_r,1}^p, \ldots, a_{n_r,j}^p, \ldots, a_{n_r,J}^p\}$, where $a_{n_r,j}^p = 1$ indicates the selection of server $j$, and $a_{n_r,j}^p = 0$ otherwise indicates non-selection. Let $x_j^p(t)$ be defined as the population share, which is the fraction of the population $p$ associating with a central server $j$ during a discrete time interval $t$, where $\sum_{j=1}^J x_j^p(t) = 1$. Therefore, the population state of the population $p$ is represented as $\mathbf{x}^p(t) = [x_1^p(t), \ldots, x_j^p(t), \ldots, x_J^p(t)]$.

*Utility.* The utility conferred upon the remaining clients is formulated as the differential between the rewards $b_j^p(t)$ obtained from participation in the unlearning process and the concomitant communication $C_{cm}^j(t)$, computation $C_{cp}^p$, and storage $C_{cs}^p$ resource costs incurred.

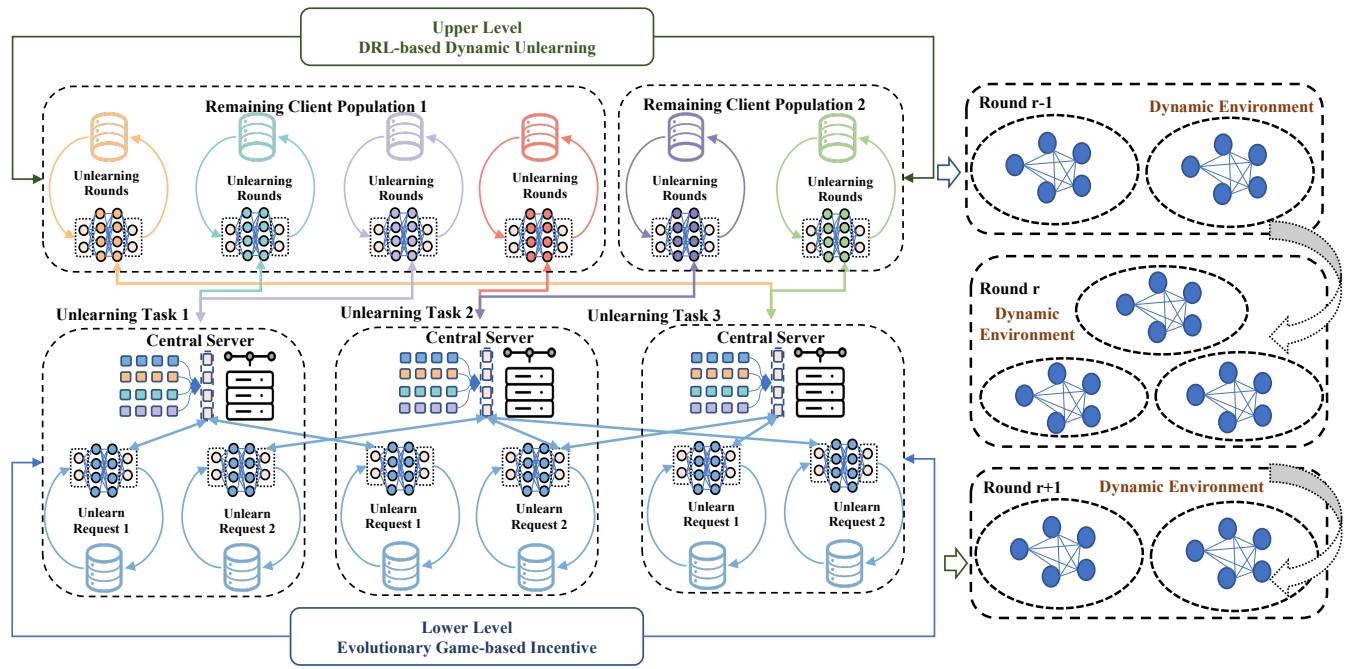

**Figure 2: Adaptive incentive and retraining framework**

## 4.2 Utility and Replicator Dynamics

*Reward.* Referring to [15, 16], the rewards, $b_j^p(t)$, conferred upon clients from population $p$ associating a central server $j$ over $R$ unlearning rounds are expressed as $b_j^p(t) = \psi_j \frac{x_j^p(t)D_p}{\sum_{p=1}^{P} x_j^p(t)D_p} + F_j^u + F_j^c$, where $\psi_j$ represents the rewards distributed by central server $j$ to incentivize remaining clients' participation in the unlearning process based on their data contribution $\frac{x_j^p(t)D_p}{\sum_{p=1}^{P} x_j^p(t)D_p}$. Additionally, $F_j^u$ is the incentives obtained by a subset of unlearning clients from original FL tasks, while $F_j^c$ denotes the penalty imposed upon unlearning clients that breach the agreement and withdraw prematurely.

*Cost.* The communication costs $C_{cm}^j(t)$, following [17], incurred by clients in the process of sharing model updates with a central server $j$ within a time period $t$ are mathematically expressed as $C_{cm}^j(t) = \frac{\zeta_j}{1-\phi}(\sum_{p=1}^{P} x_j^p(t))^2$, where $\zeta_j$ represents the congestion coefficient, determined by the communication resources of the central server $j$. The term $1 - \phi$ accounts for losses incurred during transmission, while the quadratic expression $\left(\sum_{p=1}^{P} x_j^p(t)\right)^2$ delineates the usage profile aggregated across populations for a central server [17].

The computational costs $C_{cp}^p$ associated with client calibration across $R$ unlearning rounds, referring to [18], can be represented as $C_{cp}^p = \frac{\tau_p}{2}\alpha_p \eta f_p^2 d_p$, where $\tau_p$ signifies the number of local calibrating rounds during the $r$th unlearning round. The term $\frac{\alpha_p}{2}$ denotes the effective capacitance parameter associated with the computing chipset. Additionally, $\eta$ represents the unit cost of energy consumption, $f_p^2$ characterizes the computational capability, determined by the central processing unit, and $d_p$ indicates the quantity of calibrating data samples.

The costs associated with storage, designated $C_{cs}^p$, which pertain to the preservation of historical updates to facilitate unlearning, are expressed as $C_{cs}^p = \tau_s \frac{S_p}{1-\phi}c_s$, where $\tau_s$ represents the unlearning rounds for which model updates are stored, $S_p$ denotes the sizes of historical updates, $1 - \phi$ accounts for storage loss, and $c_s$ is the unit cost of storage.

*Utility.* Define the utility, denoted as $u_j^p(t)$, for clients within population $p$ associating with central server $j$ with the aim of maximizing unlearning benefits as $u_j^p(t) = \mathcal{U}(b_j^p(t) - C_{cm}^j(t) - C_{cp}^p - C_{cs}^p)$. Here, $\mathcal{U}$ encapsulates the linear utility function, while $b_j^p$, $C_{cm}^j(t)$, $C_{cp}^p$, and $C_{cs}^p$ respectively denote the rewards and the communication, computation, and storage costs incurred. Further, let the average utility for remaining clients, distributed across $J$ central servers, be expressed as $\bar{u}^p(t) = \sum_{j=1}^{J} x_j^p(t)u_j^p$.

*Replicator Dynamics:* Given the constrained rewards and the potential for remaining clients to associate with alternative central servers to maximize utilities, we employ replicator dynamics [19] to formulate the dynamic process, which is expressed as $\dot{x}_j^p(t) = \delta x_j^p(t)(u_j^p(t) - \bar{u}^p(t))$, where $\delta$ denotes the learning rate, serving to govern the strategic adjustments of the clients.

## 4.3 Evolutionary Equilibrium

Attaining an equilibrium regardless of initial conditions, such that $\dot{x}_j^p(t) = 0$ for all time periods after realizing the initial equilibrium, is imperative in the evolutionary game. The forthcoming analysis delineates the proof pertaining to the existence, uniqueness, and stability of this evolutionary equilibrium.

**Existence.** For all $v \in \mathcal{J}$, consider the first-order derivative of $d\dot{x}_j^p(t)$ as $\frac{d\dot{x}_j^p(t)}{dx_v} = \delta\left[\frac{dx_j^p(t)}{dx_v}(u_j^p(t) - \bar{u}^p(t)) + x_j^p(t)(\frac{du_j^p(t)}{dx_v} - \frac{d\bar{u}^p(t)}{dx_v})\right]$, where $\frac{d\bar{u}^p(t)}{dx_v}) = \sum_{j=1}^J(\frac{dx_j^p(t)}{dx_v}u_j^p(t) + x_j^p(t)\frac{du_j^p(t)}{dx_v})$, and $\frac{du_j^p(t)}{dx_v} = \psi_j\left[\frac{dx_j^p(t)}{dx_v}\frac{D_p}{\sum_{p=1}^P x_j^p(t)D_p} - \frac{x_j^p(t)D_p^2}{(\sum_{p=1}^P x_j^p(t)D_p)^2}\right] - \frac{2\zeta_j}{1-\phi} \times$ $(\sum_{p=1}^P x_j^p(t))\frac{dx_j^p(t)}{dx_v}$. Given that $|\frac{d\bar{u}^p(t)}{dx_v})|$ is bounded, it follows that both $|\frac{du_j^p(t)}{dx_v}|$ and $|\frac{d\dot{x}_j^p(t)}{dx_v}|$ are likewise bounded.

**Uniqueness.** Given that $|\frac{d\dot{x}_j^p(t)}{dx_v}|$ is bounded, there exists a constant $x_j^p$ situated between $x_{j_1}^p(t) \in \mathbf{x}^p(t)$ and $x_{j_2}^p(t) \in \mathbf{x}^p(t)$, for which $|\frac{\dot{x}_{j_1}^p(t) - \dot{x}_{j_2}^p(t)}{x_{j_1}^p(t) - x_{j_2}^p(t)}| = \frac{d\dot{x}_j^p(t)}{dx_v}$, can be expressed as $|\dot{x}_{j1}^p(t) - \dot{x}_{j2}^p(t)| \le \max\{|\frac{d\dot{x}_j^p(t)}{dx_v}|\}|x_{j_1}^p(t) - x_{j_2}^p(t)|, \forall x_{j_1}^p(t), x_{j_2}^p(t) \in \mathbf{x}^p(t), \forall p \in \mathcal{P}, \forall t$. This thereby suggests that the replicator dynamics possess a unique solution regardless of the initial conditions.

**Stability.** The Lyapunov function assists in determining whether an equilibrium point of a dynamic system is stable [15]. When the system state is at an equilibrium, the change or derivative of the Lyapunov function with respect to time should be non-positive. Let us consider the Lyapunov function $\mathcal{L}(\mathbf{x}^p(t)) = (\sum_{j=1}^J \sum_{p=1}^P x_j^p(t))^2$. The first-order derivative of $\mathcal{L}(\mathbf{x}^p(t))$ with respect to $t$ can be formulated as $\frac{\mathcal{L}(dx^p(t))}{dt} = 2(\sum_{j=1}^J \sum_{p=1}^P x_j^p(t))(\sum_{j=1}^J \sum_{p=1}^P \dot{x}_j^p(t))$, where $\sum_{j=1}^J \sum_{p=1}^P x_j^p(t) = P$. Thus, for stability, per the Lyapunov conditions, it is requisite that $\sum_{j=1}^J \sum_{p=1}^P \dot{x}_j^p(t) = 0, \forall t$.

## 5 UPPER-LEVEL DYNAMIC UNLEARNING

Although the above method can incentivize the remaining clients to participate in the unlearning process, the heterogeneity among them results in suboptimal performance of the FL unlearning models. Existing solutions such as FedEraser [4] and RapidRetrain [5] opt for randomly selecting a subset of remaining clients to engage in the unlearning process. However, these approaches exhibit diminished performance due to the non-IID nature of local data on the remaining clients.

An experiment was conducted to evaluate unlearning accuracy and F1 scores of Membership Inference Attacks (MIAs) [20] on non-IID data utilizing the MNIST, FashionMNIST, and CIFAR-10 datasets over 40 unlearning rounds, comparing the efficacy of FedAvg, Fed-Eraser, and K-Center methods, as depicted in Figure 3. Initially, FedAvg executes the FL process to derive the global model, denoted as $\widetilde{\mathbf{w}}^0$, which serves as the starting point for the unlearning process. FedEraser randomly selects 10 clients $n_1^l \in \mathcal{N}\backslash n_u, l \in \{1, \ldots, 10\}$ from the set of $|\mathcal{N}\backslash n_u|$ remaining clients to engage in the unlearning of the global model across three datasets. K-Center necessitates the remaining clients $\mathcal{N}\backslash n_u$ to download the global model

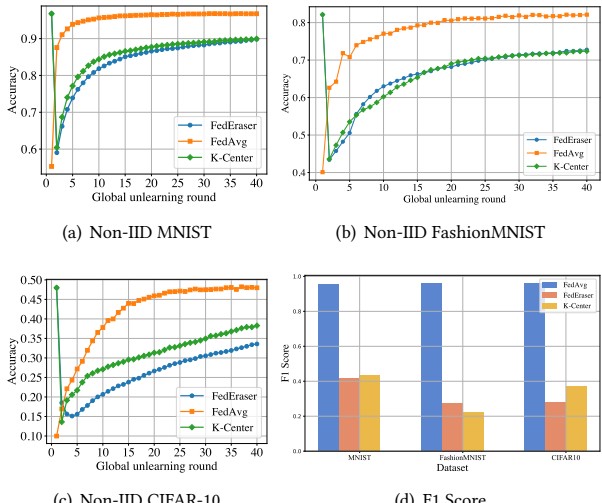

(a) Non-IID MNIST    (b) Non-IID FashionMNIST

(c) Non-IID CIFAR-10    (d) F1 Score

**Figure 3: Unlearning on non-IID data**

$\widetilde{\mathbf{w}}^0$ followed by conducting one unlearning epoch of Stochastic Gradient Descent (SGD) based on local data, yielding model updates $\{\widetilde{\mathbf{w}}_1^1, \ldots, \widetilde{\mathbf{w}}_{|\mathcal{N}\backslash n_u|}^1\}$. Thereafter, K-center clusters the remaining clients $\mathcal{N}\backslash n_u$ into 10 groups according to $\{\widetilde{\mathbf{w}}_1^1, \ldots, \widetilde{\mathbf{w}}_{|\mathcal{N}\backslash n_u|}^1\}$, subsequently selecting one client randomly from each group to participate in the unlearning process. MIAs are employed to ascertain whether specific test data was utilized during an FL model's training, thereby serving as a metric to gauge the residual information retained in the global model. The experimental configurations, including the FL model structure, datasets, and non-IID settings, are thoroughly delineated in Section 6.

As observed from the Figure 3, K-center surpasses FedEraser in performance on non-IID MNIST and non-IID FashionMNIST, and especially on non-IID CIFAR-10. Furthermore, the F1 scores pertaining to MIAs on three datasets, when utilizing both FedEraser and K-center, exhibit an approximate equivalency, showcasing a reduction of at least 45% compared to those obtained via FedAvg. Consequently, enhancing the performance of federated unlearning on non-IID data is feasible by carefully selecting remaining clients to participate in the unlearning process.

### 5.1 System Model

As depicted on the right side of Figure 2, we introduce a DRL-based mechanism for the selection of remaining clients, aimed at facilitating a dynamic unlearning process, drawing inspiration from [21]. Given that there are $\mathcal{N}_r = \mathcal{N}\backslash n_u$ remaining clients, the proposed mechanism employs a double Deep Q-network (DQN) to dynamically select $L$ remaining clients $n_r^l, l \in \{1, \ldots |\mathcal{N}_r|\}$ within each unlearning round to compute local calibration.

*State.* The state of the DQN is described by the cosine similarities between the model updates of the remaining clients $n_r^l, l \in \{1, \ldots, |\mathcal{N}_r|\}$ and the latest unlearning global model $\widetilde{\mathbf{w}}^r$. In the $r$th unlearning round, prior to the unlearning process, each client $n_r^l$ is obliged to conduct local calibration to compute the model

updates $\widetilde{\mathbf{w}}_{n_r^l}^r$ and transmit $\widetilde{\mathbf{w}}_{n_r^l}^r$ to the central server, which are utilized to construct the initial state. Subsequently, the cosine similarity can be computed as $\text{cos\_sim}(\widetilde{\mathbf{w}}_{n_r^l}^r, \widetilde{\mathbf{w}}^r) = \frac{\widetilde{\mathbf{w}}_{n_r^l}^r \cdot \widetilde{\mathbf{w}}^r}{\|\widetilde{\mathbf{w}}_{n_r^l}^r\|\|\widetilde{\mathbf{w}}^r\|} = \frac{\sum_{i=1}^{d} \widetilde{\mathbf{w}}_{n_r^l,i}^r \widetilde{\mathbf{w}}_i^r}{\sqrt{\sum_{i=1}^{d}(\widetilde{\mathbf{w}}_{n_r^l,i}^r)^2}\sqrt{\sum_{i=1}^{d}(\widetilde{\mathbf{w}}_i^r)^2}}$. Consequently, the state is represented as $\mathbf{s}^r = \{\text{cos\_sim}(\widetilde{\mathbf{w}}_{n_r^1}^r, \widetilde{\mathbf{w}}^r), \dots, \text{cos\_sim}(\widetilde{\mathbf{w}}_{n_r^{|\mathcal{N}_r|}}^r, \widetilde{\mathbf{w}}^r)\}$, which is a vector of cosine similarities.

*Action.* At the $r$th unlearning round, given the current state from the remaining clients and the latest global model, the central server selects a subset of $L$ remaining clients from the total $|\mathcal{N}_r|$ remaining clients to initiate the unlearning process. The action can be expressed as the selection, or lack thereof, of $|\mathcal{N}_r|$ remaining clients, formalized as $\mathbf{a}^r = \{a_i^r\}_{i=1}^{|\mathcal{N}_r|}$, where $a_i^r \in \{0,1\}$ indicates whether client $i$ is selected (1) or not (0) at the $r$th round. However, the complexity of this selection becomes exorbitant, equal to $C_{|\mathcal{N}_r|}^L$, particularly as the number of remaining clients enlarges. We discuss the solution in Section 5.3.

*Reward.* Given that the central server can gather states from $L$ selected clients, the accuracy, assessed via data hosted on the central server, can be utilized as a reward component to enhance performance. This is mathematically represented as $\mathbf{r}^t = \xi^{\text{cur\_acc}-\text{tar\_acc}} - 1$, where the reward $\mathbf{r}^t$ is correlated with both the current accuracy (cur\_acc) and the target accuracy (tar\_acc), and $\xi$ is a positive constant to control the sensitivity of the reward with respect to the discrepancy. Specially, $\xi$ is strategically chosen to ensure that the rewards are scaled appropriately to facilitate efficient unlearning during the training of the DQN.

### 5.2 Workflow

The workflow of DQN-based dynamic selection of remaining clients includes the following steps.

**0) Construction of the Environment.** All remaining clients retrieve the most recent unlearning model $\widetilde{\mathbf{w}}^1$ from the central server, subsequently executing a single unlearning round to acquire the model updates $\widetilde{\mathbf{w}}_{n_1^l}^1$. Following this, the updates are transmitted to the central server. The central server aggregates model updates $\widetilde{\mathbf{w}}_{n_1^l}^1$ from all remaining clients, subsequently computing the cosine similarity between $\widetilde{\mathbf{w}}_{n_1^l}^1$ and $\widetilde{\mathbf{w}}^1$. Consequently, it derives the initial state, $\mathbf{s}^1$, utilizing the cosine similarity, $\text{cos\_sim}(\widetilde{\mathbf{w}}_{n_r^l}^r, \widetilde{\mathbf{w}}^r)$, where $l \in \{1, \dots, |\mathcal{N}_r|\}$.

**1) Dynamic Selection.** In the $r$th unlearning round, the central server decides the selection of $L$ participants from the remaining clients and receives model updates from the selected clients $\widetilde{\mathbf{w}}_{n_r^{l'}}^r, l' \in \{1, \dots, L\}$ for aggregation and distribution. The client selection in the $r$th unlearning round involves computing the value function $Q_p(\mathbf{s}^t, a; \mathbf{w}_d)$, where $a$ is the client selection and $\mathbf{w}_d$ are the parameters of the policy network. There are two networks in the DQN: the policy and target networks, which share the same architecture. The central server employs an $\epsilon$-greedy strategy to select actions from the action space. Specifically, it computes the value function $Q_p(\mathbf{s}^t, a; \mathbf{w}_d)$ and selects the top-$L$ values' corresponding indices as the newly selected clients in a probability $1 - \epsilon$, performing local calibration and transmission subsequently. The central server utilizes the next states from the selected clients to replace the subset of corresponding previous states.

The update of the value function typically follows the temporal-difference (TD) error rule. The formula for updating the Q-values in DQN is often given by $Q_t(\mathbf{s}^t, a) = \text{reward} + \gamma \max_{a'} Q_p(\mathbf{s}^{t+1}, a'; \mathbf{w}_t)$, where $Q_t$ is the target Q-value, which the policy network aims to approach. $\gamma$ is the discount factor, determining the present value of future rewards. $\max_{a'} Q_p(\mathbf{s}^{t+1}, a'; \mathbf{w}_t)$ is the maximum estimated future reward when transitioning to the next state $\mathbf{s}^{t+1}$. $\mathbf{w}_t$ are the parameters of the target network. The loss $L$ for updating the policy network parameters $\mathbf{w}_d$ can then be computed using mean squared error between the Q-target and the current Q-value approximation as $L(\mathbf{w}_d) = \mathbb{E}\left[\left(Q_t(\mathbf{s}^t, a) - Q_p(\mathbf{s}^t, a; \mathbf{w}_d)\right)^2\right]$, which is then used to perform a gradient descent update on the policy network parameters.

**2) Local Calibration and Transmission.** During the $r$th unlearning round, the selected clients retrieve the latest unlearning model $\widetilde{\mathbf{w}}^r$, calculate the local model updates $\widetilde{\mathbf{w}}_{n_r^{l'}}^r$ utilizing their respective local data $D_{n_r^{l'}}$, and subsequently transmit the results to the central server for aggregation.

**3) Unlearning Aggregation.** The central server collects local model updates $\widetilde{\mathbf{w}}_{n_r^{l''}}^r$, and utilizes them for calibration via the step length and direction, assessed layer by layer, as elaborated in Section 3.2. Subsequently, the new global unlearning model $\widetilde{\mathbf{w}}^{r+1}$ is aggregated and distributed to the selected clients for the subsequent $(r+1)$th unlearning round.

The aforementioned steps 1-3 will be iteratively executed for $R$ unlearning rounds to systematically mitigate the data influences from specific clients.

### 5.3 Discussion

Given that the action spaces enlarge substantially as the number of remaining clients increase, various methodologies, notably Principal Component Analysis (PCA) [21], hierarchical frameworks [15], and sharding [22], which have found extensive application in FL, can also be judiciously deployed in the federated unlearning process to mitigate complexity and curtail dimensional size.

## 6 EXPERIMENTAL RESULTS

### 6.1 Lower-Level Evolutionary Game

In our experimental setup, we engage 90 remaining clients, each possessing varying data quantities. The clients are partitioned into three distinct populations, wherein each population encompasses 30 clients, and they are categorized based on their respective data quantities for unlearning, ranging between [80,120] data samples per client. The computation cost, $C_{cp}^p$, and the storage cost, $C_{cs}^p$, are set at 0.1. The learning rate, denoted as $\delta$, is established at 0.001. Rewards, $\psi_j$, distributed by each central server $j$ are variably set between [100,300]. The congestion coefficient, $\zeta_j$, is situated between [10,20]. Meanwhile, the transmission loss, $\phi$, is designated at 0.5. Incentives $F_j^u$ and $F_j^c$ emanating from remaining clients are

respectively fixed between [80,120] and [30,40]. Initially, one-third of the clients from each population are assigned evenly across three central servers. The aforementioned parameters and configurations are inspired and adjusted based on previous works [15, 16]. Except where explicitly stated, the simulation parameters adhere to the aforementioned configurations.

Figure 4 illustrates the evolution of population utilities. The method proposed in this study is compared with random, round-robin, and greedy strategies. Specifically, under the random policy, the remaining clients are assigned without any strategic consideration. Conversely, the round-robin policy assigns clients in a cyclical order, while the greedy policy allocates clients based on the available rewards. These policies are examined and adapted following the methodologies discussed in [23].

Figure 4 (a) illustrates the total utilities over 300 time periods, contrasting our proposed method with random, round-robin, and greedy strategies. All methodologies attain a stable equilibrium within 100 time periods, thereby substantiating the stability outlined in Section 4.3. Remarkably, our evolutionary game-based strategy emerges superior, achieving at the best total utility compared with random, round-robin, and greedy strategies. This can be attributed to its capability to adeptly navigate the dynamics of rewards and costs across central servers, thereby facilitating remaining clients from various populations in identifying and associating with a central server for retraining tasks, all while maximizing utility.

Figure 4 (b) illustrates the total utilities over four distinct learning rates, juxtaposing our proposed method with random, round-robin, and greedy strategies, focusing on the time periods at 50 to examine the evolutionary equilibrium under diverse conditions. It is discernible from the figure that the trends of the four strategies uniformly display a decrease in total utilities with increasing learning rates. This indicates that a lower learning rate necessitates a more extended period to achieve the evolutionary equilibrium. This phenomenon is attributable to the learning rates modulating the speed of the replicator dynamics. Specifically, a lower learning rate results in a decelerated pace of the replicator dynamics, demanding more time to diminish to zero and thereby achieve equilibrium.

Figure 4 (c) displays the average utilities associated with our proposed method, compared to random, round-robin, and greedy strategies, given a time period of 50 and a learning rate of 0.001, across a range of remaining client quantities from 10 to 70. Notably, the figure reveals a consistent decreasing trend in average utilities across all four methods as the number of clients escalates. This decrease in average utilities, despite the efforts of strategies to guide clients towards optimal central servers for retraining tasks, can be attributed to the dilution of reward pools; as the client count augments, the rewards must be disseminated among a larger pool of participants, thus diminishing the average utility per client.

Figures 4 (d), (e), and (f) elucidates the accuracy resultant from 40 unlearning rounds, which are inherently associated with population shares generated by the evolutionary game under four distinct populations [0.1, 0.4, 0.5], [0.4, 0.4, 0.2], [0.8, 0.1, 0.1], and [0.0, 0.4, 0.6]. The figure compares our proposed method with random, round-robin, and greedy strategies, upon a non-iid dataset comprising MNIST [24], Fashion-MNIST [25], and CIFAR-10 (CIFAR) [26]. The architecture of FL models, trained on MNIST, Fashion-MNIST, and

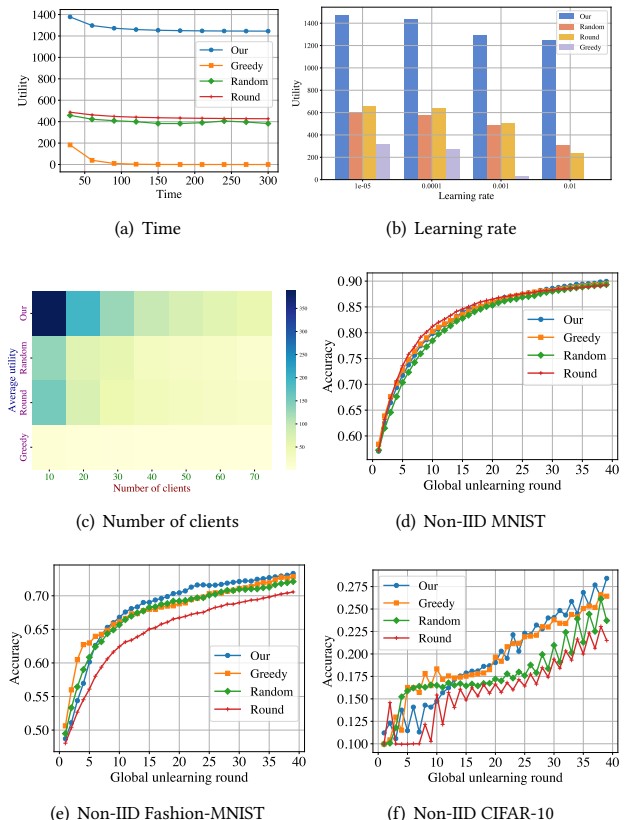

(a) Time

(b) Learning rate

(c) Number of clients

(d) Non-IID MNIST

(e) Non-IID Fashion-MNIST

(f) Non-IID CIFAR-10

**Figure 4: Evolution of population utilities and accuracy**

CIFAR-10 refers to FedEraser [4]. Observations derived from the figures demonstrate a notable consistency in the accuracy achieved by the four comparative methods across 40 unlearning rounds. Although the proposed method can attract a greater share of the population to participate in the unlearning process, the calibration accuracy during the unlearning process is perceptibly influenced by the heterogeneity of the remaining clients' local data. Consequently, it necessitates the development of a methodology adept at mitigating such heterogeneity, thereby enhancing accuracy.

## 6.2 Upper-Level Dynamic Retraining

The upper-level dynamic retraining process is executed on Ubuntu 16.04.7 LTS, equipped with 8 cores, 64GB of memory, and an NVIDIA RTX 3090 GPU. A total of 100 clients are engaged in the experiment, and a subset of these, excluding unlearning clients, are selected in the unlearning process to reduce waiting time for model aggregation [3]. The local data for unlearning on each client is a non-IID setting, in which each client retained $\frac{total\_samples}{total\_clients}$ samples following [21]. Within this data subset of each client, 80% originated from a dominant class, with the remaining 20% pertaining to alternative classes. During each unlearning round, 10 of the remaining clients are selected to participate in federated unlearning. The local unlearning round was set to 5 compared with the local training round 10, the global unlearning round to 40, with a local batch size of

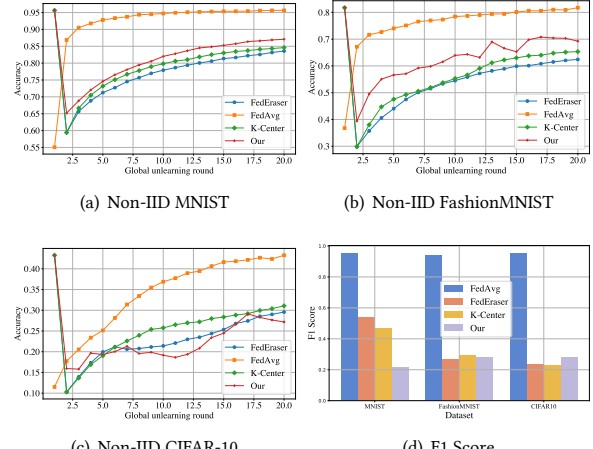

(a) Non-IID MNIST      (b) Non-IID FashionMNIST

(c) Non-IID CIFAR-10      (d) F1 Score

**Figure 5: Unlearning on non-IID data with three datasets**

64, a local learning rate of 0.005, and the utilization of Stochastic Gradient Descent (SGD) as the optimization algorithm. Our Deep Q-Network (DQN) model, implemented using PyTorch, comprises a straightforward three-layer architecture, designed to process states and actions related to 99 remaining clients. It features an input layer, two hidden layers with 8 and 256 neurons respectively, both featuring ReLU activations, and an output layer projecting 256 neurons to 99 Q-values without additional activation. The model leverages a replay buffer for experience replay during training, storing up to 1,000 tuples of experiences and sampling in batches of 8 to stabilize learning. Utilizing the mean squared error loss as its loss function and Adam optimizer with a learning rate of 0.001, the DQN balances swift and stable learning. Moreover, an $\epsilon$-greedy strategy ($\epsilon$ = 0.1) is employed to facilitate exploration during training. Unless otherwise specified, the parameters are established in accordance with the methodologies outlined in the aforementioned studies [4, 21].

As illustrated in Figure 5, the proposed method is compared with FedAvg, FedEraser, and K-Center across MNIST, FashionMNIST, and CIFAR-10 datasets to ascertain accuracy across 20 unlearning rounds. In this comparison, FedAvg serves as the baseline, derived from the FL training process. Conversely, FedEraser employs a strategy of randomly selecting participants from the pool of remaining clients to engage in the unlearning process. Meanwhile, K-Center organizes the remaining clients into 10 clusters, randomly selecting a single client from each group to participate in the unlearning process. As depicted in Figures 5 (a), (b), and (c), our methodology surpasses both FedEraser and K-Center in terms of performance on the MNIST and FashionMNIST datasets, while exhibiting diminished performance on CIFAR-10. Nonetheless, all three methods yield a uniformly low accuracy, approximately 30%, when applied to the CIFAR-10 dataset. Furthermore, it is observable from Figure 5 (d) that the F1 score of the three methodologies can diminish to approximately 0.3, illustratively contrasting the unlearning performance relative to FedAvg. The performance across the three methods demonstrates a near uniformity.

## 7 CONCLUSION

We propose a two-level incentive and unlearning mechanism, designed to encourage the participation of remaining clients in the unlearning process and to dynamically select the number of remaining clients with unbalanced local data, thus optimizing the equilibrium and performance efficacy. Compared to benchmark strategies, the proposed mechanisms can maximize utility within the low level and achieve dynamic remaining client selection without retraining from scratch. We envision this work as an initiative to integrate networking into federated unlearning, achieving an effective interplay. In future work, we will explore methods to erase historical data effects from both participating clients and servers, examine decentralized unlearning algorithms, and integrate differential privacy, employing Gaussian noises to mitigate data effects.

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
