# OpenReview forum: "Incentive and Dynamic Client Selection for Federated Unlearning"
_ACM.org/TheWebConf/2024/Conference — TheWebConf24_

### Official Review · Reviewer_mUvV · 2023-11-22

**Novelty:** 6
**Technical Quality:** 5

**Review:**

In this manuscript, the authors present an approach for federated unlearning by incentive and dynamic client selection. Their approach utilize evolutionary game theory to model the dynamic participation process，and use the DRL to dynamically select the remaining clients. This solves the problem of residual client utilization, as well as non-IID data on the client. The idea of this manuscript is very innovative, and the questions raised are also practical problems that need to be solved urgently. In this paper, the effectiveness of the method is proved by various experiments .However, there are still some problems that can be improved:

* There are fewer baselines, only three. Authors can add 2-3 up-to-date methods of federated unlearning.
* In Figure 5c, the accuracy curve on the CIFAR-10 dataset is very strange. Further analysis by the authors is needed
* The font size in the experimental result graph is too small to be enlarged.

**Questions:**

In Figure 2, why is the line between the upper level and the lower level like this? Is there any pattern? I don't seem to see an explanation in the manuscript.

**Ethics Review Description:**

No ethics issue.

**Reviewer Confidence:**

2: The reviewer is willing to defend the evaluation, but it is likely that the reviewer did not understand parts of the paper

**Scope:**

3: The work is somewhat relevant to the Web and to the track, and is of narrow interest to a sub-community

---

### Official Review · Reviewer_dREy · 2023-11-23

**Novelty:** 4
**Technical Quality:** 3

**Review:**

The paper proposes an approach to client selection for federated unlearning. The suggested solution utilizes evolutionary game theory to model the dynamic participation process and employs deep reinforcement learning to dynamically select the remaining clients to join the unlearning process. Overall, the paper is easy to follow, with insights well-presented. However, several concerns arise after reading the manuscript, outlined below.

* Motivation for the evolutionary game

Although the paper mentions that the "unlearning process imposes an additional burden by requiring the computational and communication resources of the remaining clients," it is still unclear why the proposed rewarding evolutionary game is necessary for federated learning (FL) unlearning. In Section 4.2, the utility definition is discussed, suggesting that utility represents the actual benefit for each remaining client based on rewards and costs, with rewards linked to data contribution. Such a game strategy may incentivize clients to pursue higher data contribution, potentially leading to a scenario where clients with lower costs benefit more from the utility. In a worst-case scenario, clients with less data and higher communication, computational, or storage costs may benefit less and may eventually withdraw from the game. Please clarify whether such a situation is possible and explain the measures taken to prevent it, ensuring that disadvantaged clients always have the opportunity to contribute data to the FL model.

* Fairness

Concerning dynamic unlearning, the remaining clients are carefully selected using a reinforcement learning-assisted approach, implying that each remaining client may have an unequal probability of participating in the unlearning process. Please provide clarification or more discussion on this concern, emphasizing the need to consider the fairness of each client in the selection process.

**Questions:**

1. Why the rewarding evolutionary game is necessary?
2. How the fairness of disadvantaged clients is guaranteed?

**Reviewer Confidence:**

3: The reviewer is confident but not certain that the evaluation is correct

**Scope:**

3: The work is somewhat relevant to the Web and to the track, and is of narrow interest to a sub-community

---

### Official Review · Reviewer_a5wP · 2023-11-23

**Novelty:** 4
**Technical Quality:** 5

**Review:**

This paper explores the timely and critical approach of federated unlearning, especially given the involvement of confidential and secure data in federated learning. The paper proposes a cluster-based joint incentive and unlearning approach for federated unlearning. The technical details and design structure are thoroughly described.

However, there are some questions requiring further clarification.

How does the central server determine the successful completion of unlearning?

What is the communication cost and impact on the two-level approach?

Are there a sufficient number of rounds to facilitate the unlearning process at both local and global levels?

What are the time units for Figure 4C?

What benefits arise from having identical samples among clients within a population, despite the initial notion that client data differs?

What implications does unlearning have on model performance and accuracy?

How does the process of unlearning influence model stability and convergence in subsequent training?

**Questions:**

This paper explores the timely and critical approach of federated unlearning, especially given the involvement of confidential and secure data in federated learning.

There are some questions requiring further clarification:

1) How does the central server determine the successful completion of unlearning?

2) What is the communication cost and impact on the two-level approach?

3) Are there a sufficient number of rounds to facilitate the unlearning process at both local and global levels?

4) What are the time units for Figure 4C?

5) What benefits arise from having identical samples among clients within a population, despite the initial notion that client data differs?

6) What implications does unlearning have on model performance and accuracy?

7) How does the process of unlearning influence model stability and convergence in subsequent training?

**Reviewer Confidence:**

2: The reviewer is willing to defend the evaluation, but it is likely that the reviewer did not understand parts of the paper

**Scope:**

3: The work is somewhat relevant to the Web and to the track, and is of narrow interest to a sub-community

---

### Decision · Program_Chairs · 2024-01-22

**Decision:**

Accept

**Comment:**

The paper proposes an approach to client selection for federated unlearning. The proposed solution utilizes evolutionary game theory to model the dynamic participation process and employs deep reinforcement learning to dynamically select the remaining clients to join the unlearning process. This is an interesting line of work with a number of noteworthy aspects. However, the reviewers also raised concerns, which if addressed can improve the paper.

 *Pros:*
 1. Detailed technical descriptions; thorough design structure.
 2. Easy to follow; well-presented insights.
 3. Innovative approach; practical problem-solving; effectiveness demonstrated by various experiments.

 *Cons:*
 1. Concerns about fairness in client selection
 2. Lack of baselines; more recent methods need to be compared against (e.g., RapidRetrain, INFOCOM'22)
 3. Lack of discussion about the central server's role in unlearning, communication costs, and the impact of identical samples among clients.